# Subjective Self-Assessment After Microneedling Radiofrequency or Platelet-Rich Plasma and Fibrin Treatments and Motivating Factors for Their Use

**DOI:** 10.3390/healthcare13060634

**Published:** 2025-03-14

**Authors:** Dajana Malarz, Łukasz Czyżewski, Dorota Olczak-Kowalczyk

**Affiliations:** 1Department of Geriatric Nursing, Medical University of Warsaw, 02-091 Warsaw, Poland; dajana.orlowska.m@gmail.com; 2Department of Pediatric Dentistry, Medical University of Warsaw, 02-091 Warsaw, Poland; dorota.olczak-kowalczyk@wum.edu.pl

**Keywords:** dermatology, female, medicine, motivational interviewing, self-assessment

## Abstract

**Background/Objectives**: Today, aesthetic medicine is becoming increasingly common, with the demand for rejuvenating treatments continuously growing. These procedures are gaining more interest from patients who struggle to accept their aesthetic imperfections. There is a need to understand patients’ subjective evaluations of the effects of aesthetic medicine procedures and the factors that motivate them to undergo invasive treatments. Understanding these aspects allows for better tailoring of therapy to patients’ needs, increasing their satisfaction and quality of life. Additionally, it enables the identification of psychological and social factors influencing their decisions, supporting a more conscious and responsible approach to aesthetic medicine. **Methods**: This study included 82 women who had undergone treatments with platelet-rich plasma and fibrin (50%) or microneedling radiofrequency (50%), with an average age of 44.3 years. The research questionnaire consisted of 12 original questions. This study was conducted from January to March 2024. Over 90% of the women had been undergoing treatments for at least several months. **Results**: Participants reported a significantly improved assessment of their appearance after the procedure, with an average increase of 2.72 points on a scale from 1 to 10 (M = 2.72; SD = 1.66) (*p* < 0.001). Women from rural areas significantly more often (29.6%) cited appearance-related complexes as a reason for the procedure compared to women from cities (3.6%) and large cities (3.7%) (*p* = 0.003). Over 80% reported an increase in self-confidence due to the procedures they underwent. **Conclusions**: The analysis showed a significant difference in the assessment of appearance before and after the procedure; participants rated their appearance more favorably after the procedure. The motivating factors for women to undergo aesthetic medicine procedures include the desire to look younger and more attractive. The marital status of the respondents did not influence motivating factors.

## 1. Introduction

Today, aesthetic medicine is becoming increasingly common, and the demand for rejuvenating treatments continues to rise. These procedures are gaining more interest from patients who struggle to accept their aesthetic imperfections. Aesthetic medicine procedures, both invasive, such as needle mesotherapy, microneedling radiofrequency, botulinum toxin, and tissue stimulators, as well as non-invasive procedures like oxygen infusion, lifting facial massages, and needle-free mesotherapy, form an integral part of this medical field. The current trend is the intelligent selection of therapies for comprehensive facial rejuvenation, such as combining microneedling radiofrequency, platelet-rich plasma, and fibrin with fillers. The use of radiofrequency in aesthetic medicine continues to evolve and has remained a key element for many years. Studies confirm the effectiveness of this method in reducing wrinkles and fat tissue [1]. Platelet-rich plasma and fibrin are current therapeutic methods due to their proven clinical efficacy, inducing extracellular matrix remodeling, stimulating the proliferation of skin fibroblasts, and collagen synthesis [2]. These procedures not only correct physical defects, slow down the aging process, and improve appearance but also support the enhancement of patients’ psychological and emotional well-being. Despite the prevalence of these procedures, there is a need to understand patients’ subjective assessments of the effects of aesthetic medicine treatments and the factors motivating them to undergo invasive procedures.

Aesthetic medicine is not only about appearance but also about promoting healthy behaviors, including proper diet, physical exercise, and regular skincare routines [3]. Daily skincare using appropriate cosmetics and routine care procedures helps maintain skin in good condition, preventing problems such as acne, discoloration, and wrinkles. Regular cleansing, moisturizing, and using SPF sunscreen protect the skin from harmful external factors and delay the aging process.

Although beauty standards change, youth remains a constant feature of attractiveness, and the fear of aging significantly impacts women’s self-esteem. The rise of social media has amplified women’s dissatisfaction with their appearance, driving them to strive for unattainable standards [4,5]. The decision to undergo invasive aesthetic medicine procedures may be motivated by various factors that stem from both personal needs and expectations, as well as social and cultural influences. These include increasing the sense of physical attractiveness, boosting self-esteem and confidence, eliminating physical imperfections such as facial asymmetry, visible wrinkles, or skin blemishes, striving to maintain a youthful appearance, and societal pressure regarding physical appearance to meet standards or align with current trends [6]. Subjective self-assessment by patients after undergoing aesthetic medicine procedures plays a key role in their experience and satisfaction with the results. In addition to external evaluations, patients also assess their internal feelings, emotions, and changes in self-perception. These procedures can help patients feel more satisfied with their appearance and foster a more positive outlook on themselves, which can have a profound impact on their well-being and quality of life. However, subjective self-assessment may also change depending on individual patient expectations and their perception of the procedure’s outcomes. Overall patient satisfaction depends not only on the results of the procedure itself but also on the price, availability, the course of the visit, and the qualifications of the staff. It is important for the patient to feel comfortable and for the procedure to be carried out professionally, in accordance with their expectations [7]. A survey conducted by the International Society of Aesthetic Plastic Surgery revealed a global increase in the use of aesthetic medicine procedures [8]. In 2015, 12.1 million procedures were performed, with the highest numbers in the United States and Brazil. Among non-surgical procedures, botulinum toxin has been the most used (since 1999), aimed at minimizing facial expression wrinkles, followed by fillers—cross-linked hyaluronic acid administered to fill nasolabial folds and enlarge lips [9].

This study aims to determine the factors motivating women to undergo aesthetic medicine procedures and to examine how women’s self-assessments change before and after injection treatments.

## 2. Materials and Methods

A questionnaire-based study was conducted among women who underwent invasive aesthetic medicine procedures. Women who did not consent to participate or who incorrectly completed the questionnaire were excluded from the study. The research questionnaire consisted of 12 original questions. The questions were multiple-choice and closed-ended. The first part of the questionnaire concerned age, type of employment, and daily skincare routines. The second part focused on the factors motivating the women to undergo aesthetic medicine procedures, the types of procedures they had received, and their assessment of their appearance before and after the treatments. The validation of the questionnaire was conducted on two levels: content validity and reliability. Content validity was assessed through consultations with experts in aesthetic medicine, who evaluated whether the questions in the questionnaire were appropriate for the studied issues. Reliability was assessed using Cronbach’s alpha coefficient, which was 0.85, indicating high internal consistency of the questions. The questionnaire was tailored to the target group, which consisted of women aged 35–55 who had undergone aesthetic procedures. The initial version of the questionnaire was tested on a small group of women, which allowed for the identification of issues related to the understanding of questions and answers, which were subsequently optimized. The participants had undergone treatments with platelet-rich plasma and fibrin (50%) or microneedling radiofrequency (50%). This study was conducted from January to March 2024. The results were subjected to descriptive and statistical analysis using the SPSS version 20.0 statistical program. A standard significance level of 0.05 was adopted to verify the results, and a *t*-test was used to analyze dependencies.

## 3. Results

The study involved 90 women, with 8 excluded due to incorrectly filled questionnaires. Ultimately, the study group consisted of 82 participants. The average age of the respondents at the time of the study was 44.33 years. Most women (78%) practiced daily skincare, including the use of creams or serums.

Table 1 presents the characteristics of the study group. The largest group of participants were women aged 35–40 years. Intellectual work slightly predominated over physical labor (52.4%). Nearly half of the participants were married (48.8%).

Over 90% of the women have been undergoing treatments for at least several months. The most frequently chosen procedures (Table 2) are platelet-rich plasma and fibrin (71.9%) and needle mesotherapy (35.4%).

The motivating factors for women to undergo invasive aesthetic medicine procedures are presented in Table 3. The primary reasons for utilizing aesthetic medicine are the desire to look younger (51.2%) and to increase attractiveness (25.3%). More than half of the women do not feel pressured by society or the media to undergo injection treatments, nor do they compare themselves to celebrities.

Table 4 illustrates the subjective self-assessment of appearance before and after the procedure, as well as whether participants perceived any positive effects post-treatment. Over 60% of the women rated their appearance as 5 or below prior to the procedure, while only 8.5% did so after. Only a small percentage (7%) reported not noticing any positive changes in their appearance post-procedure.

Table 5 illustrates the participants’ perception of changes in their work, relationship with their partner, and social interactions after the procedure. Most women reported no significant changes in these areas, while nearly 82% experienced increased self-confidence following the procedure.

The paired *t*-test demonstrated a statistically significant difference in self-assessment of appearance before and after the procedure (*t*(81) = −14.86; *p* < 0.001). Participants reported a significantly improved assessment of their appearance after the procedure, with an average increase of 2.72 points on a scale from 1 to 10 (M = 2.72; SD = 1.66)—Table 6.

Table 7 shows a statistically significant difference in the frequency of appearance-related complexes as a reason for undergoing invasive aesthetic medicine procedures among women from different locations (Chi-square(2) = 11.43; *p* = 0.003). Women from rural areas significantly more often (29.6%) cited appearance-related complexes as a reason for the procedure compared to women from cities (3.6%) and large cities (3.7%).

Statistical tests were conducted to determine whether there were statistically significant differences among groups of women regarding the factors motivating them to undergo aesthetic medicine procedures. It was found that there is no statistically significant difference among women of different ages in terms of whether a woman: (1) feels pressured by society or the media to undergo invasive procedures (Chi-square(3) = 5.99; *p* = 0.112); (2) compares her appearance to celebrities/people on TV (Chi-square(3) = 0.69; *p* = 0.877); (3) desires to look more attractive (Chi-square(3) = 0.87; *p* = 0.832); (4) desires to look younger (Chi-square(3) = 0.54; *p* = 0.910); (5) indicates peer pressure from friends (Chi-square(3) = 1.06; *p* = 0.786); (6) indicates an appearance-related complex (Chi-square(3) = 3.53; *p* = 0.317); or (7) indicates difficulty accepting aging (Chi-square(3) = 2.44; *p* = 0.487).

There is no statistically significant difference among women with different types of jobs in terms of whether a woman: (1) feels pressured by society or the media to undergo invasive procedures (Chi-square(1) = 0.30; *p* = 0.586); (2) compares her appearance to celebrities/people on TV (Chi-square(1) = 0.14; *p* = 0.710); (3) desires to look more attractive (Chi-square(1) = 0.69; *p* = 0.407); (4) desires to look younger (Chi-square(1) = 0.21; *p* = 0.650); (5) indicates peer pressure from friends (Chi-square(1) = 1.75; *p* = 0.186); (6) indicates an appearance-related complex (Chi-square(1) = 2.30; *p* = 0.129); or (7) indicates difficulty accepting aging (Chi-square(1) = 1.29; *p* = 0.255).

There is no statistically significant difference among women of different marital statuses in terms of whether a woman: (1) feels pressured by society or the media to undergo invasive procedures (Chi-square(4) = 6.18; *p* = 0.186); (2) compares her appearance to celebrities/people on TV (Chi-square(4) = 2.59; *p* = 0.628); (3) desires to look more attractive (Chi-square(4) = 2.21; *p* = 0.698); (4) desires to look younger (Chi-square(4) = 5.29; *p* = 0.259); (5) indicates peer pressure from friends (Chi-square(4) = 2.23; *p* = 0.694); (6) indicates an appearance-related complex (Chi-square(4) = 3.57; *p* = 0.467); or (7) indicates difficulty accepting aging (Chi-square(4) = 5.96; *p* = 0.203).

There is no statistically significant difference among women from different-sized towns in terms of whether a woman: (1) feels pressured by society or the media to undergo invasive procedures (Chi-square(2) = 1.35; *p* = 0.508); (2) compares her appearance to celebrities/people on TV (Chi-square(2) = 0.77; *p* = 0.681); (3) desires to look more attractive (Chi-square(2) = 5.64; *p* = 0.060); (4) desires to look younger (Chi-square(2) = 1.49; *p* = 0.476); (5) indicates peer pressure from friends (Chi-square(2) = 0.34; *p* = 0.842); or (6) indicates difficulty accepting aging (Chi-square(2) = 3.37; *p* = 0.186).

Tests were conducted to determine whether there were statistically significant differences in the increase in self-assessment levels (calculated as the difference between post-procedure and pre-procedure ratings) among different groups of women. Using ANOVA, it was found that there are no statistically significant differences: (1) among women of different age groups in the increase in self-assessment levels (F(3, 78) = 0.18; *p* = 0.910); (2) in the increase in self-assessment levels among women of different marital statuses (F(4, 77) = 1.25; *p* = 0.296); (3) in the increase in self-assessment levels among women from different sized towns (F(2, 79) = 0.24; *p* = 0.786); (4) in the increase in self-assessment levels among women with different experience levels in undergoing procedures (F(4, 77) = 0.46; *p* = 0.766); (5) in the increase in self-assessment levels between women who have undergone different types of procedures (*t*(80) = 0.07; *p* = 0.947); (6) in the increase in self-assessment levels between women who have undergone needle mesotherapy and those who have not (*t*(80) = 0.26; *p* = 0.797); (7) in the increase in self-assessment levels between women who have undergone microneedling mesotherapy and those who have not (*t*(80) = −1.63; *p* = 0.106); (8) in the increase in self-assessment levels between women who have undergone laser treatments and those who have not (*t*(80) = 1.05; *p* = 0.295); (9) in the increase in self-assessment levels between women who have undergone PRP and fibrin treatments and those who have not (*t*(80) = 0.66; *p* = 0.513); (10) in the increase in self-assessment levels between women who have undergone cross-linked hyaluronic acid treatments and those who have not (*t*(80) = 0.42; *p* = 0.675); (11) in the increase in self-assessment levels between women who have undergone tissue stimulator treatments and those who have not (*t*(80) = −1.41; *p* = 0.162); (12) in the increase in self-assessment levels between women who have undergone botulinum toxin treatments and those who have not (*t*(80) = 0.17; *p* = 0.867).

## 4. Discussion

Subjective self-assessment, or how an individual evaluates themselves and their traits, is a complex issue influenced by factors such as culture, social media, interpersonal relationships, and life experiences. In the context of aesthetic medicine, subjective self-assessment can be impacted both positively and negatively. Today, where external appearance is increasingly valued, understanding the impact of aesthetic procedures on self-esteem is crucial for both doctors and psychologists.

Aesthetic medicine, once associated primarily with individuals over 45, is undergoing a significant transformation. Nowadays, younger individuals are increasingly seeking preventative treatments. Studies show that the largest group of patients are women over 40 [10], but younger people are also frequenting aesthetic clinics. In this study, most participants were aged 35–40, with an average age of 44.3 years.

Research conducted by the Polish Society of Aesthetic Medicine in 2017 showed that Polish women most frequently opt for aesthetic treatments such as botulinum toxin (29%), fillers (28%), and microneedling (18%). Despite the increasing number of people choosing these treatments, studies indicate that the majority of women do not admit to undergoing aesthetic procedures within their close social circles. This is often due to the fear of judgment from others and the stigma that can still be associated with aesthetic medicine, especially in cultures where a natural appearance is often considered more authentic. Many women fear that their decision to undergo such procedures may be perceived as a sign of a lack of acceptance for their bodies. However, it is worth noting that this phenomenon may change, as in recent years, aesthetic medicine has become increasingly socially accepted, with more and more discussions framing it as a way to improve quality of life and self-esteem. This shift is largely due to the growing social awareness and changing attitudes towards aging and personal appearance care [11]. Among the procedures analyzed, the most chosen were platelet-rich plasma and fibrin (71.9%) and microneedling (35.4%). Botulinum toxin and cross-linked hyaluronic acid ranked 4th and 6th, respectively, in terms of frequency of choice. The American Society of Aesthetic Plastic Surgery notes a growing trend towards procedures that promote a more natural appearance, which may explain the increased choice of revitalizing treatments [9].

Women’s motivations for undergoing aesthetic procedures often include a desire to look younger and enhance their attractiveness. Kurczabińska and Nienańska [12] and Maisel et al. [13] confirm that the primary reasons are visible signs of aging and the need to improve sexual attractiveness. It is crucial to establish patient expectations before starting therapy and tailor the treatments individually, which can positively affect the patient’s quality of life [14]. In the research Kurczabińska-Luboń and Nieznańska on the use of minimally invasive aesthetic procedures, the largest group consisted of women aged 46–55. In this age group, the most commonly chosen procedures were those using dermal fillers, with 52% of respondents opting for them. Mesotherapy, one of the more popular treatments, was chosen by 12% of women in this group. The main motivation behind the decision to undergo treatments was visible signs of skin aging and the desire to improve physical attractiveness. As many as 24% of respondents stated that they were dissatisfied with their appearance and were looking for ways to improve their well-being. The vast majority of respondents expected the procedure to have a positive effect on their mood and self-esteem. After the procedures, the research results showed that most of the participants indeed experienced an improvement in their well-being and noticed an increase in self-confidence. Increased attractiveness, although one of the main goals, turned out to be just one element of a broader process of improving quality of life, which also included the psychological aspect of feelings related to external appearance. This confirms that minimally invasive aesthetic procedures not only affect appearance but also improve overall satisfaction with oneself and quality of life [12].

In the study, 41.5% of participants felt societal or media pressure to undergo aesthetic treatments. Kozaczyńska [15] highlights that only a small percentage of women choose treatments due to external pressure. Aesthetic medicine is undeniably linked to social media. The American Society of Aesthetic Plastic Surgery noted that with the advent of social media, the number of aesthetic procedures increased by 466% [16]. Studies conducted by Abbasini et al. emphasize the significant influence of platforms like Instagram on potential patients, increasing their desire to undergo aesthetic procedures. Researchers noted that dentists participating in their study mostly use social media to advertise their services, gaining a new patient base. Due to their easy accessibility, constant updates, and globalization, social media has a huge impact on people’s attitudes and self-esteem [17]. American society invests substantial amounts in aesthetic procedures, which may influence self-confidence and social acceptance [18,19]. Mojs draws special attention to this—although aesthetic medicine was previously associated with financial success, many people now undergo procedures even when they do not have sufficient financial resources. The willingness to go into debt for the sake of a better appearance highlights the determination of those undergoing treatments and the growing social pressure to look young. Advertisements circulating on television and the internet emphasize that women want to be young, as it is inseparably associated with beauty, strength, success, and openness. Over the years, the belief in the unattractiveness of one’s own body has dramatically increased worldwide, along with the belief that aesthetic procedures have a positive impact on one’s mental state and various life areas, improving well-being, providing a sense of inner comfort, and increasing attractiveness, which is highly sought after in today’s world [20]. In the study, 82% of participants reported increased self-confidence following treatments.

The subjective evaluation of appearance before treatments varied widely. The largest group (23.2%) rated their appearance as 5 on a 10-point scale. No one rated their appearance as 9, and only one person rated it as 10. After the procedure, the largest group (34.1%) rated their appearance as 8, 21 people rated it as 9, and four people rated it as 10. Over 90% of participants noticed positive changes following the procedure. However, there are limitations to these results, as beauty is a completely subjective concept, and no standardized or widely accepted tools are available [21]. There is also a lack of clear data on how the quality of life for patients undergoing aesthetic procedures changes, with only references to improvements being correlated with plastic surgery [22].

Glen T. Porter and Francin B. Quinn demonstrate that attractive people tend to find partners more easily, advance more quickly, are served faster, and are perceived as nicer [23]. Research by Galęba found that 29% of individuals felt that aesthetic procedures helped them find a partner, although it did not significantly affect their chances of obtaining a higher-paying job [3]. In the study, most respondents reported no changes in their work, relationships, or social interactions following aesthetic procedures. One-quarter of respondents noted positive changes in their work.

The limitations of the study include several important factors that should be considered when interpreting the results. Firstly, the sample size was relatively small, which may affect the generalizability of the findings to a larger population. Additionally, this study was conducted in a specific geographical location, which limits the representativeness of the results, especially regarding regional and socio-cultural differences that may influence decisions to undergo aesthetic procedures. Another limitation is the lack of a control group, which prevents comparison with individuals who did not undergo treatments. There is also a potential risk that external factors, such as changes in social media, beauty trends, or marketing campaigns, may have influenced the results, particularly in terms of participants’ perceptions and motivations. Finally, the lack of long-term follow-up in the study is a limitation, as changes in self-perception and satisfaction levels after the procedures may evolve over time.

## 5. Conclusions

The analysis showed a significant difference in self-assessment of appearance before and after the procedure, with participants rating their appearance on average 2.72 points higher on a 10-point scale after the procedure. The main motivation for women to undergo aesthetic procedures is the desire to look younger and more attractive. There was no statistically significant difference in self-esteem increase related to age. Marital status did not impact the motivating factors.

## Figures and Tables

**Table 1 healthcare-13-00634-t001:** Characteristics of the Study Group.

Parameter	Range	n (%)
Age	35–40	24 (29.4%)
	41–45	20 (24.4%)
	46–50	19 (23.2%)
	51–55	19 (23.2%)
Type of work	Mental work	43 (52.4%)
	Physical work	39 (47.6%)
Marital status	Single	14 (17.1%)
	Married	40 (48.8%)
	Divorced	13 (15.8%)
	Widow	5 (6.1%)
	Relationship	10 (8.2%)
Number of kids	1	25 (30.5%)
	2	26 (31.7%)
	3 or more	13 (15.9%)
	Without	18 (21.9%)
Place of residence	Village/small town 27 (32.9%)	
	City	28 (34.2%)
	Large city	27 (32.9%)

**Table 2 healthcare-13-00634-t002:** Types of invasive procedures performer duration since the patient started undergoing aesthetic medicine procedures.

Types and Time of Invasive Procedures	n (%)
Needle mesotherapy	29 (35.4%)
Microneedling mesotherapy	26 (31.7%)
Laser treatments	19 (23.2%)
Platelet-rich plasma and fibrin	59 (71.9%)
Cross-linked hyaluronic acid treatments	16 (19.5%)
Tissue stimulators	14 (17.1%)
Botulinum toxin	21 (25.6%)
First time	8 (10%)
Few months	18 (22%)
1 to 2 years	16 (20%)
3 to 10 years	31 (38%)
More than 10 years	9 (11%)

**Table 3 healthcare-13-00634-t003:** Motivating Factors for Women to Undergo Invasive Aesthetic Medicine Procedures.

Parameter		n (%)
Factors		
	I wanted to look more attractive	29 (25.3%)
	I wanted to look younger	42 (51.2%)
	My friends encouraged me	7 (8.5%)
	I had a complex related to my appearance	10 (8.2%)
	I couldn’t come to terms with aging	17 (20.1%)
Comparing oneself to famous people		
	Yes	34 (41.5%)
	No	48 (58.5%)
Pressure from society/media		
	Yes	28 (34.1%)
	No	54 (65.9%)

**Table 4 healthcare-13-00634-t004:** Subjective Self-Assessment of Appearance Before and After the Procedure, and Perceived Effect of the Procedure.

Parameter		n (%)
Appearance rating before		
	1	2 (2.4%)
	2	6 (7.3%)
	3	12 (14.6%)
	4	12 (14.6%)
	5	19 (23.2%)
	6	11 (13.4%)
	7	16 (19.6%)
	8	3 (3.7%)
	9	0 (0%)
	10	1 (1.2%)
Appearance rating after		
	1	0 (0%)
	2	1 (1.2%)
	3	1 (1.2%)
	4	2 (2.4%)
	5	3 (3.7%)
	6	7 (8.5%)
	7	15 (18.3%)
	8	28 (34.1%)
	9	21 (25.6%)
	10	4 (4.9%)
Effect of the procedure		
	Appears better	76 (92.7%)
	No, she did not notice any improvement	6 (7.3%)

**Table 5 healthcare-13-00634-t005:** Perception of Self-Confidence and Changes in Work, Relationship, and Social Interactions.

Parameter		n (%)
Perception of Self-Confidence		
	Yes	37 (45.1%)
	Rather yes	30 (36.6%)
	I do not know	12 (14.6%)
	No	4 (4.8%)
Changes in work		
	Promotion	0 (0%)
	Colleagues/bosses are more favorable	21 (25.6%)
	No change	61 (74.4%)
	Negative changes	0 (0%)
Changes in relationship		
	Partner considers me more attractive	18 (21.95%)
	No change	44 (53.7%)
	Negative change	2 (2.4%)
	Not applicable	18 (21.95%)
Changes in social interactions		
	Improved	14 (17.1%)
	No change	60 (73.2%)
	Negative effects-jealousy from colleagues/sarcastic comments	8 (9.7%)

**Table 6 healthcare-13-00634-t006:** Dependency of Self-Assessment of Appearance on Procedure Completion.

	How Would You Rate Your Appearance on a Scale from 1 to 10 Before Undergoing the Procedure?	How Would You Rate Your Appearance on a Scale from 1 to 10 After Undergoing the Procedure?
N	82	82
Average	4.94	7.66
Median	5	8
Mode	5	8
Standard deviation	1.84	1.53
Minimum	1	2
Maximum	10	10

**Table 7 healthcare-13-00634-t007:** Dependency of Residence Location on Motivating Factors for Women to Undergo Aesthetic Medicine Procedures.

Parameter		I Had a Complex Related to My Appearance	
		No	Yes
Place of residence			
	Village	19 (70.4%)	8 (29.6%)
	City	27 (96.4%)	1 (3.6%)
	Large city	26 (96.3)	1 (3.7%)

## Data Availability

The data presented in this study are available on request from the corresponding author.

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
