# Peer review of "Subjective Self-Assessment After Microneedling Radiofrequency or Platelet-Rich Plasma and Fibrin Treatments and Motivating Factors for Their Use"

_healthcare, 2025, doi:10.3390/healthcare13060634_

Round 1
Reviewer 1 Report
Comments and Suggestions for Authors
Thank you for submitting your manuscript.This study investigates a relevant topic in aesthetic medicine, and your findings on patient motivations and self-perception are interesting. However, we have some suggestions to improve the clarity and impact of your manuscript:
1. The current title is quite broad. Consider revising it to be more specific to your target population and the interventions studied.
2. Please revise the abstract for a concise description of the study design and the main findings.
3. While the introduction provides relevant background, it could benefit from some restructuring to improve the flow and focus. Consider adding more citations to support some of the statements and ensure proper attribution.
4: The methods section would be strengthened by providing a more detailed description of the questionnaire used in the study. Include information on how it was developed, validated, and implemented.
5: The discussion could be more thorough and engage more deeply with existing literature. Consider expanding the discussion on each specific domain (motivating factors, self-assessment, etc.) and comparing your findings with those from other studies.
6: Please address the study's limitations explicitly in the Discussion section. This includes the relatively small sample size, the specific geographic location and its potential influence on the results, the lack of a control group, and the absence of long-term follow-up data.
7: Correctly format all references.
Comments on the Quality of English Language
It is okay.
Author Response
Comments 1: The current title is quite broad. Consider revising it to be more specific to your target population and the interbentions studied. |
||||
Response 1: Thank you for pointing this out. We agree with this comment. Therefore, we have changed title to more specific. Line 2-3 “Subjective Self-Assessment After Microneedling Radiofrequency or Platelet-Rich Plasma and Fibrin Treatments and Motivating Factors for Their Use”
|
||||
Comments 5: The discussion could be more thorogh an engage more deeply with existing literaturę. Consider expanding the discussion on each specific domain (motivating factors, self-assessment, etc. ) and comparing your findings with those from other studies. Response 5: Agree. We have, accordingly, modified discussion to emphasize this point. We add sentences in line 362-364, line 375-390, line 394-401, line 403-414. We added 5 well-chosen references.
Research conducted by the Polish Society of Aesthetic Medicine in 2017 showed that Polish women most frequently opt for aesthetic treatments such as botulinum toxin (29%), fillers (28%), and microneedling (18%). Despite the increasing number of people choosing these treatments, studies indicate that the majority of women do not admit to undergoing aesthetic procedures within their close social circles. This is often due to the fear of judgment from others and the stigma that can still be associated with aesthetic medicine, especially in cultures where a natural appearance is often considered more authentic. Many women fear that their decision to undergo such procedures may be perceived as a sign of lack of acceptance for their bodies. However, it is worth noting that this phenomenon may change, as in recent years, aesthetic medicine has become increasingly socially accepted, with more and more discussions framing it as a way to improve quality of life and self-esteem. This shift is largely due to the growing social awareness and changing attitudes towards aging and personal appearance care. In the research Kurczabińska- Luboń and Nieznańska on the use of minimally invasive aesthetic procedures, the largest group consisted of women aged 46-55. In this age group, the most commonly chosen procedures were those using dermal fillers, with 52% of respondents opting for them. Mesotherapy, one of the more popular treatments, was chosen by 12% of women in this group.The main motivation behind the decision to undergo treatments was visible signs of skin aging and the desire to improve physical attractiveness. As many as 24% of respondents stated that they were dissatisfied with their appearance and were looking for ways to improve their well-being. The vast majority of respondents expected the procedure to have a positive effect on their mood and self-esteem. After the procedures, the research results showed that most of the participants indeed experienced an improvement in their well-being and noticed an increase in self-confidence. Increased attractiveness, although one of the main goals, turned out to be just one element of a broader process of improving quality of life, which also included the psychological aspect of feelings related to external appearance. This confirms that minimally invasive aesthetic procedures not only affect appearance but also improve overall satisfaction with oneself and quality of life.
Aesthetic medicine is undeniably linked to social media. The American Society of Aesthetic Plastic Surgery noted that with the advent of social media, the number of aesthetic procedures increased by 466%. Studies conducted by Abbasini et al. emphasize the significant influence of platforms like Instagram on potential patients, increasing their desire to undergo aesthetic procedures. Researchers noted that dentists participating in their study mostly use social media to advertise their services, gaining a new patient base. Due to their easy accessibility, constant updates, and globalization, social media has a huge impact on people's attitudes and self-esteem. Mojs draws special attention to this – although aesthetic medicine was previously associated with financial success, many people now undergo procedures even when they do not have sufficient financial resources. The willingness to go into debt for the sake of a better appearance highlights the determination of those undergoing treatments and the growing social pressure to look young. Advertisements circulating on television and the internet emphasize that women want to be young, as it is inseparably associated with beauty, strength, success, and openness. Over the years, the belief in the unattractiveness of one's own body has dramatically increased worldwide, along with the belief that aesthetic procedures have a positive impact on one's mental state and various life areas, improving well-being, providing a sense of inner comfort, and increasing attractiveness, which is highly sought after in today’s world.
Comments 6: Please address the study’s limitations explicity in the Discussion section. This includes the relatively small sample size, the specific geografic location and its potential influence on the results, the lack of a control gropu, and the absence of long- term follow-up data.
|
||||
Response 6: Thank you for pointing this out. We agree with this comment. Therefore, we have add it to manuscript in page 9. Line 432-443 “The limitations of the study include several important factors that should be considered when interpreting the results. Firstly, the sample size was relatively small, which may affect the generalizability of the findings to a larger population. Additionally, the study was conducted in a specific geographical location, which limits the representativeness of the results, especially regarding regional and socio-cultural differences that may influence decisions to undergo aesthetic procedures. Another limitation is the lack of a control group, which prevents comparison with individuals who did not undergo treatments. There is also a potential risk that external factors, such as changes in social media, beauty trends, or marketing campaigns, may have influenced the results, particularly in terms of participants’ perceptions and motivations. Finally, the lack of long-term follow-up in the study is a limitation, as changes in self-perception and satisfaction levels after the procedures may evolve over time.” |
Comments 7: Correctly format all references.
Response 7: Thank you for pointing this out. We have carefully corrected all references according to the journal guidelines.
Reviewer 2 Report
Comments and Suggestions for Authors
I’ve had the opportunity to read the manuscript entitled Subjective Self-Assessment of Female Patients After Selected Aesthetic Medicine Procedures and Motivating Factors for Their Use.
The study aims to investigate the self-assessment trough a research questionnaire of 12 original questions and specific motivations that justify the willingness of undergoing invasive aesthetic procedures of a selected cohort of 82 female patients that received different aesthetic treatments. The results indicate that participants experienced an average increase of 2.72 points in their self-assessed appearance on a scale of 1 to 10. The main motivating factors identified were the desire to look younger and enhance attractiveness. The authors also evaluated potential societal and media pressure that was found to be insignificant.
Overall, I find the article interesting for the readers of the journal. To date the growing request for aesthetic procedures in globally increasing and becoming more and more common also in general population. The main objective of this study to investigate the self-evaluation of the results of these treatments made by the patients and the motivations that may incentivize patients to undergo also invasive aesthetic procedures could provide an interesting point of view for medical professionals.
Regarding the specific section of the manuscript, I’d like to offer the following feedback to enhance the quality of the manuscript further.
Materials and Methods
- The sample size is relatively small, and it can limit the statistical power and the generalizability of the findings, I think that this point should be better disclosed in the limitation section
Discussion
- The authors correctly reported that results in subjective self-assessment may introduce bias, as participants might overestimate or underestimate their improvements due to psychological (and other) factors. I think that this section should be expanded with a more in-depth analysis regarding the potential consequences of this bias.
- Despite the authors cited some similar studies in the discussion section, the manuscript can be improved with a more specific analysis and comparison with prior studies including studies that report neutral or negative effects of aesthetic procedures.
Author Response
Comments 1: The sample size is relatively small, and i tan limit the statistical power and the generalizability of the findings, i think that this point should be better disclosed in the limitation section.
Response 1: Thank you for pointing this out. We agree with this comment. Therefore, we have add it to manuscript in page 9. Line 432-443
“The limitations of the study include several important factors that should be considered when interpreting the results. Firstly, the sample size was relatively small, which may affect the generalizability of the findings to a larger population. Additionally, the study was conducted in a specific geographical location, which limits the representativeness of the results, especially regarding regional and socio-cultural differences that may influence decisions to undergo aesthetic procedures. Another limitation is the lack of a control group, which prevents comparison with individuals who did not undergo treatments. There is also a potential risk that external factors, such as changes in social media, beauty trends, or marketing campaigns, may have influenced the results, particularly in terms of participants’ perceptions and motivations. Finally, the lack of long-term follow-up in the study is a limitation, as changes in self-perception and satisfaction levels after the procedures may evolve over time.”
Comments 2: Discussion - The authors correctly reported that results in subjective self-assessment may introduce bias, as participants might overestimate or underestimate their improvements due to psychological (and other) factors. I think that this section should be expanded with a more in-depth analysis regarding the potential consequences of this bias.
Response 2: Thank you for pointing this out. We agree with this comment. Therefore, we have add it to manuscript in page 2. Line 63-66 and 79-83.
Although beauty standards change, youth remains a constant feature of attractiveness, and the fear of aging significantly impacts women's self-esteem. The rise of social media has amplified women's dissatisfaction with their appearance, driving them to strive for unattainable standards [4,5].
Overall patient satisfaction depends not only on the results of the procedure itself but also on the price, availability, the course of the visit, and the qualifications of the staff. It is important for the patient to feel comfortable and for the procedure to be carried out professionally, in accordance with their expectations [7].
Comments 3: Discussion – Despite the authors cited some similar studies in the discussion section, the manuscript can be improved with a more specific analysis and comparison with prior studies including studies that report neutral or negative effects of aesthetic procedures
Rosponse 3: Agree. We have, accordingly, modified discussion to emphasize this point. We add sentences in line 362-364, line 375-390, line 394-401, line 403-414. We added 5 well-chosen references.
Research conducted by the Polish Society of Aesthetic Medicine in 2017 showed that Polish women most frequently opt for aesthetic treatments such as botulinum toxin (29%), fillers (28%), and microneedling (18%). Despite the increasing number of people choosing these treatments, studies indicate that the majority of women do not admit to undergoing aesthetic procedures within their close social circles. This is often due to the fear of judgment from others and the stigma that can still be associated with aesthetic medicine, especially in cultures where a natural appearance is often considered more authentic. Many women fear that their decision to undergo such procedures may be perceived as a sign of lack of acceptance for their bodies. However, it is worth noting that this phenomenon may change, as in recent years, aesthetic medicine has become increasingly socially accepted, with more and more discussions framing it as a way to improve quality of life and self-esteem. This shift is largely due to the growing social awareness and changing attitudes towards aging and personal appearance care.
In the research Kurczabińska- Luboń and Nieznańska on the use of minimally invasive aesthetic procedures, the largest group consisted of women aged 46-55. In this age group, the most commonly chosen procedures were those using dermal fillers, with 52% of respondents opting for them. Mesotherapy, one of the more popular treatments, was chosen by 12% of women in this group.The main motivation behind the decision to undergo treatments was visible signs of skin aging and the desire to improve physical attractiveness. As many as 24% of respondents stated that they were dissatisfied with their appearance and were looking for ways to improve their well-being. The vast majority of respondents expected the procedure to have a positive effect on their mood and self-esteem. After the procedures, the research results showed that most of the participants indeed experienced an improvement in their well-being and noticed an increase in self-confidence. Increased attractiveness, although one of the main goals, turned out to be just one element of a broader process of improving quality of life, which also included the psychological aspect of feelings related to external appearance. This confirms that minimally invasive aesthetic procedures not only affect appearance but also improve overall satisfaction with oneself and quality of life.
Aesthetic medicine is undeniably linked to social media. The American Society of Aesthetic Plastic Surgery noted that with the advent of social media, the number of aesthetic procedures increased by 466%. Studies conducted by Abbasini et al. emphasize the significant influence of platforms like Instagram on potential patients, increasing their desire to undergo aesthetic procedures. Researchers noted that dentists participating in their study mostly use social media to advertise their services, gaining a new patient base. Due to their easy accessibility, constant updates, and globalization, social media has a huge impact on people's attitudes and self-esteem.
Mojs draws special attention to this – although aesthetic medicine was previously associated with financial success, many people now undergo procedures even when they do not have sufficient financial resources. The willingness to go into debt for the sake of a better appearance highlights the determination of those undergoing treatments and the growing social pressure to look young. Advertisements circulating on television and the internet emphasize that women want to be young, as it is inseparably associated with beauty, strength, success, and openness. Over the years, the belief in the unattractiveness of one's own body has dramatically increased worldwide, along with the belief that aesthetic procedures have a positive impact on one's mental state and various life areas, improving well-being, providing a sense of inner comfort, and increasing attractiveness, which is highly sought after in today’s world.

Round 2
Reviewer 1 Report
Comments and Suggestions for Authors
The manuscript has improved